# Isolation and Characterization of the First Temperate Virus Infecting *Psychrobacillus* from Marine Sediments

**DOI:** 10.3390/v14010108

**Published:** 2022-01-08

**Authors:** Wang Liu, Xiaowei Zheng, Xin Dai, Zhenfeng Zhang, Wenyan Zhang, Tian Xiao, Li Huang

**Affiliations:** 1State Key Laboratory of Microbial Resources, Institute of Microbiology, Chinese Academy of Sciences, Beijing 100101, China; liuwang17@mails.ucas.ac.cn (W.L.); zhengxw@im.ac.cn (X.Z.); daix1970@163.com (X.D.); zhangzf@im.ac.cn (Z.Z.); 2College of Life Science, University of Chinese Academy of Sciences, Beijing 100049, China; 3Key Laboratory for Marine Ecology and Environmental Sciences, Institute of Oceanology, Chinese Academy of Sciences, Qingdao 266071, China; zhangwy@qdio.ac.cn (W.Z.); txiao@qdio.ac.cn (T.X.); 4Center for Ocean Mega-Science, Chinese Academy of Sciences, Qingdao 266071, China; 5Laboratory for Marine Ecology and Environmental Science, Qingdao National Laboratory for Marine Science and Technology, Qingdao 266273, China

**Keywords:** *Psychrobacillus*, bacteriophage, marine sediments, *Myoviridae*

## Abstract

Viruses are far more abundant than cellular microorganisms in the marine ecosystem. However, very few viruses have so far been isolated from marine sediments, especially hydrothermal vent sediments, hindering the understanding of the biology and ecological functions of these tiny organisms. Here, we report the isolation and characterization of a temperate bacteriophage, named PVJ1, which infects *Psychrobacillus* from a hydrothermal vent field in Okinawa Trough. PVJ1 belongs to the *Myoviridae* family of the order *Caudovirales*. The tailed phage possesses a 53,187 bp linear dsDNA genome, with 84 ORFs encoding structural proteins, genome replication, host lysis, etc. in a modular pattern. The phage genome is integrated into the host chromosome near the 3′-end of *deoD*, a gene encoding purine nucleoside phosphorylase (PNP). The phage integration does not appear to disrupt the function of PNP. The phage DNA is packaged by the headful mechanism. Release of PVJ1 from the host cell was drastically enhanced by treatment with mitomycin C. Phages encoding an MCP sharing significant similarity (≥70% identical amino acids) with that of PVJ1 are widespread in diverse environments, including marine and freshwater sediments, soils, artificial ecosystems, and animal intestines, and primarily infect Firmicutes. These results are valuable to the understanding of the lifestyle and host interactions of bacterial viruses at the bottom of the ocean.

## 1. Introduction

Viruses are the most abundant biological entities in the ocean, and those in marine sediments are especially abundant and extraordinarily diverse [1,2,3,4]. Virus-like particles (VLPs) in some marine sediments were found to exceed 10^9^/mL, which is three orders of magnitude higher than VLPs in seawater at the same depth [5]. VLPs of different morphologies, predominated by the head-tail shape, were observed under an electron microscope in the eluates of marine sediment samples [1,2,3], but the vast majority of the VLP sequences retrieved from the metagenomic data of the sediment samples showed no significant matches to known sequences [3]. Although the virosphere of marine sediments remains largely a mystery, its role in geobiochemical processes in the oceans has been increasingly recognized. Clearly, the isolation and characterization of these viruses are essential to the elucidation of their ecological functions.

Hydrothermal vents support unique ecosystems, in which considerably diverse microorganisms driving elemental cycling thrive [6]. However, fewer than a dozen of bacteriophages have been isolated from hydrothermal vents and characterized so far [7,8,9,10,11]. Four of them infect *Bacillus* or *Geobacillus*, and they are all lytic [8,9,10,11]. These phages possess head-tailed morphotypes similar to those of the majority of phages isolated from terrestrial hot springs [12]. Therefore, head-and-tailed viruses of the order *Caudovirales*, which account for most of the prokaryotic viruses [13], appear to be also well represented in the extreme environments on Earth. Metagenomic analysis shows that viruses in hydrothermal vents are predominantly lysogenic and carry auxiliary metabolic genes that function to regulate the physiological activities of the host [14]. It is suggested that proviruses contribute to the fitness of the host strains, whereas the integrated state represents a strategy employed by the viruses to survive harsh conditions [15].

Here, we report the isolation and characterization of PVJ1, a bacteriophage that infects a mesophilic *Psychrobacillus* strain, from a sediment sample taken from a hydrothermal vent field in Okinawa Trough. PVJ1 is a temperate phage of the *Myoviridae* family of the order *Caudovirales*, and capable of integrating its 53,187 bp linear dsDNA genome into the host genome at a unique site within a gene encoding purine nucleoside phosphatase. Phages encoding a MCP sharing significant similarity (≥70% identical amino acids) with that of PVJ1 specifically infected Firmicutes and are widespread in aquatic environments, soils, and animal intestines.

## 2. Materials and Methods

### 2.1. Strain Isolation

Samples used in this study were collected in July 2018, during the cruise conducted by the scientific research vessel KEXUE in Okinawa Trough (GPS coordinates: 126.55° E, 27.78° N, depth: 958 m, temperature in situ: 4 °C). The sediments were collected by TV grab (TVG) equipped on a KEXUE vessel. After removing the superficial 5 cm layer of sediments, surface sediments (upper 10 cm) were transferred in aseptic sampling bags (Haibo, Qingdao, China). Sediments were stored at −80 °C and kept on dry ice during transportation. Then, 5 g sediment samples were inoculated into marine 2216E medium [16]. After incubation at 18 °C for 1–2 weeks, the enrichment cultures were plated onto marine agar 2216E. Following further incubation at 18 °C for 1 week, colonies were picked, and isolates were purified by streaking repeatedly on 2216E plates. The 16S rRNA gene of each isolate was amplified by PCR using the primers 16S-27F/1492R (Appendix A) and sequenced. *Psychrobacillus* sp. GC2J1, the native host of bacteriophage PVJ1, was obtained.

### 2.2. Isolation and Purification of PVJ1

*Psychrobacillus* sp. GC2J1 was grown with shaking in marine 2216E medium at 28 °C to an OD_600_ of 0.4~0.6. Mitomycin C (Amresco) was added to a final concentration of 1 μg/mL to induce the replication and release of PVJ1. After incubation for 9 h, the culture was centrifuged at 10,000× *g* for 20 min at 4 °C. The supernatant was filtered through a membrane with a 0.22 μm pore size (Merck Millipore). Virus particles in the filtrate were collected by ultracentrifugation at 120,000× *g* for 1 h at 4 °C and gently resuspended in buffer SM (10 mM Tris-HCl, pH 7.5; 100 mM NaCl; 10 mM MgSO_4_). The particles were subjected to purification by cesium chloride density gradient centrifugation at 200,000× *g* for 24 h at 4 °C. The purified PVJ1 particles were resuspended in buffer SM and stored at 4 °C.

### 2.3. Transmission Electron Microscopy (TEM)

A sample (10 μL) of the phage PVJ1 was deposited on a carbon-coated 230-mesh grid, negatively stained for 1 min with 2% (*w*/*v*) uranyl acetate, and observed under a JEM-1400 transmission electron microscope (JEOL, Tokyo, Japan).

### 2.4. DNA Isolation and Genome Analysis

Purified PVJ1 particles were digested with proteinase K (0.6 mg/mL) in the presence of 2% (wt/vol) SDS at 55 °C for 3 h. NaCl and cetyltrimethylammonium bromide (CTAB) were added to 800 mM and 1%, respectively. After incubation for 10 min at 65 °C, the sample was extracted twice with phenol:chloroform:isoamylalcohol (25:24:1), and the DNA was precipitated with ethanol. Paired-end (PE) sequencing (2 × 150 bp) was conducted on an Illumina Hiseq-1500 platform at Microbial Genome Research Center, Institute of Microbiology, Chinese Academy of Sciences, Beijing, China. Raw reads were adapter trimmed and quality controlled using fastp (v0.19.4) with default parameters [17] and subsequently assembled by SPAdes 3.11 [18] with a series of kmers (i.e., 21, 31, 55, 77, 99, 121). The resulting PVJ1 genome was submitted to RAST (http://rast.nmpdr.org/ accessed on 11 December 2021) for an initial annotation [19] and further checked by BLASTp analysis against NCBI non-redundant (nr) protein database. Hypothetical proteins were also screened for distinct homologs in HHpred (https://toolkit.tuebingen.mpg.de/tools/hhpred accessed on 11 December 2021) [20]. Putative tRNA genes were predicted with tRNAscan-SE (http://lowelab.ucsc.edu/tRNAscan-SE/ accessed on 11 December 2021). Gene domain architectures were predicted with SMART (http://smart.embl-heidelberg.de accessed on 11 December 2021). Transmembrane domains and signal peptides were predicted using TMHMM (http://www.cbs.dtu/services/TMHMM/ accessed on 11 December 2021) and Signal P-3.0 Server (http://www.cbs.dtu.dk/services/SignalP-3.0/ accessed on 11 December 2021), respectively.

### 2.5. Quantification of Phage DNA by qPCR

The PVJ1 DNA was quantified by qPCR with the primer pair PVJ1-qPCR-F/R (Appendix A) targeting PVJ1 ORF81, which encodes a portal protein. To prepare a standard curve for PVJ1 DNA quantification, the PCR fragment was cloned into plasmid pEASY-T1 (TransGen Biotech, Beijing, China). The construct was then serially diluted and amplified as the known control to generate a standard curve relating the copy number of PVJ1 DNA with CT (threshold cycle) values. The standard reaction (20 μL) contained 2 μL of total DNA template, 10 μL of TB Green^TM^ Premix Ex Taq^TM^ II (Tli RNaseH Plus, Takara Bio, Shiga, Japan), 0.8 μL each of the primers (10 μM), and 6.4 μL of DNase-free water. The two-step qPCR reaction was performed on a Light Cycler fluorescence quantitative PCR instrument (Roche Life Science, Indianapolis, IN, USA) with pre-denaturation for 30 s at 95 °C, followed by 40 cycles of denaturation for 5 s at 95 °C, and annealing and extension for 30 s at 60 °C in the first step, and denaturation for 5 s at 95 °C, annealing and extension for 1 min at 60 °C, and denaturation for heating to 95 °C, and cooling to 50 °C in the second step [21].

### 2.6. Host Range Determination

The plaque assays and qPCR quantification were used to screen the phage's ability to lyse different strains. Tested strains included *Psychrobacillus* sp. GC2J1 (CGMCC 1.19103), *Psychrobacillus insolitus* DSM5 (CGMCC 1.3683), *Psychrobacillus lasiicapitis* NEAU-3TGS17^T^ (CGMCC 1.15308), *Lysinibacillus antri* SYSU K30002^T^ (CGMCC 1.13504), *Paenisporosarcina antarctica* N-05 (CGMCC 1.6503), and *Planococcus rifietoensis* N14 (CGMCC 1.8017). For plaque assays, phage suspension (10 μL) was detected on lawns of the strains grown on plates. After overnight incubation at 25 °C, plates were observed for the presence of clear or turbid plaques. For qPCR quantification, 10 μL phage suspension was added into 500 μL bacterial cultures at exponential growth phase, Mitomycin C was added after culturing to the stationary phase, and the total PVJ1 DNA before and after induction was quantified by qPCR.

### 2.7. Protein Analysis

The purified PVJ1 virions were subjected to 12% sodium dodecyl sulfate-polyacrylamide gel electrophoresis (SDS–PAGE). The gel was stained with Coomassie Brilliant Blue G250, and gel slices containing protein bands were excised. Proteins in the gel slices were digested with trypsin (12.5 ng/μL), and the resulting peptides were analyzed by matrix-assisted laser desorption ionization-time of flight (MALDI-TOF) mass spectrometry (AB 533 Sciex 5800 TOF/TOF 5800, Foster City, CA, USA) as described previously [21].

### 2.8. Atomic Force Microscopy (AFM)

The purified PVJ1 DNA molecules were dissolved in nuclease-free water, adsorbed onto the mica substrate in the presence of 5 mM Mg^2+^, and observed under an atomic force microscope (Nanoscope V Multimode, Bruker, USA). Length measurements of DNA molecules were performed manually using the software Image J (Wayne Rasband, National Institute of Health, USA). A total of 50 fields were examined.

### 2.9. Phylogenetic Analysis

The amino acid sequences of the terminase large subunits of PVJ1 and 32 other viruses with a clear classification of DNA packaging strategies [22] were aligned by Clustal W [23], and a neighbor-joining phylogenetic tree was constructed using MEGA 7.0 with a bootstrap of 1000 replicates.

Bacterial and viral proteins were downloaded from the NCBI reference sequence database (ftp://ftp.ncbi.nlm.nih.gov/genomes/refseq/bacteria/ accessed on 13 September 2021) and IMG/VR database (Integrated Microbial Genomes with Virus, https://img.jgi.doe.gov/vr/ accessed on 13 September 2021). The habitat information of these sequences was manually parsed from the NCBI and IMG/VR databases. Proteins annotated as capsid were extracted to create a local database using Diamond (V.0.9.10.111) [24]. Homologs of the major capsid protein (ORF75) of PVJ1 were extracted after aligning ORF75 against the local capsid database using “diamond blastp” (≥70% identity and ≥50% alignment, E-value of 1 × 10^−3^). The obtained capsid sequences were multi-aligned by the program MAFFT 7.455 with the “auto” algorithm [25], followed by sequence trimming using trimAl 1.4.1 in default parameters [26]. Additionally, a maximum likelihood (ML) phylogenetic tree was constructed for capsid proteins with FastTree v.2.1.10 in the WAG + GAMMA model [27]. Further, integrated prophage sequences in bacterial genomes were predicted using PHASTER [28]. Those viral sequences containing capsid clustered closely with ORF75, as observed in the ML phylogenetic tree of the capsid, were selected for synteny analysis using EasyFig 3.0.5 [29].

## 3. Results and Discussion

### 3.1. Isolation of Psychrobacillus sp. GC2J1 and Bacteriophage PVJ1

Strain GC2J1 was isolated during a survey of microorganisms in sediment samples collected from a hydrothermal vent field in Okinawa Trough. The strain grew at 4–40 °C (opt: 25–30 °C) and pH 5–12 (opt: pH 7–8) in the presence of 0–5% (*w*/*v*) NaCl (opt: 2%). GC2J1 was capable of forming endospores, a property that presumably allows the strain to withstand occasional exposure to high temperatures in the habitat. The sequence of the 16S rRNA gene of strain GC2J1 (GenBank accession no. MZ959810) is 99.9% identical to that of *Psychrobacillus lasiicapitis* NEAU-3TGS17^T^ (GenBank accession no. NR_159144), a strain isolated from the head of an ant [30]. However, digital DNA–DNA hybridization and average nucleotide identity (ANI) values for the two strains were only 36.6% and 89%, respectively, suggesting that GC2J1 is probably a new species in the genus *Psychrobacillus*. It should be noted that, although *Psychrobacillus* strains were isolated from a variety of environments, e.g., oil-contaminated soil, red fox feces, the Antarctic iceberg, etc., *Psychrobacillus* sp. GC2J1 represents the first member of the genus *Psychrobacillus* to be isolated from high-temperature habitats.

The strain denoted *Psychrobacillus* sp. GC2J1 released virus-like particles (VLPs) following treatment with mitomycin C (1 μg/mL; Figure 1A,B). We analyzed 50 VLPs by TEM to measure diameter, length, and width. VLPs possess an icosahedral head of 60 ± 2 nm in diameter and a contractile tail of 120 ± 5 and 20 ± 1 nm in length and width, respectively (Figure 1A). A cluster of terminal knobs of 20 ± 2 nm in length was observed. Based on its morphological characteristics, the bacteriophage belongs to the *Myoviridae* family of the order *Caudovirales*. We named the phage *Psychrobacillus* virus J1 (PVJ1). To demonstrate that PVJ1 indeed existed in the sediment sample from the hydrothermal vent field in Okinawa Trough, we extracted metagenomic DNA from the sample and conducted a PCR assay using a primer pair targeting PVJ1 (PVJ1-ORF75F/R, Appendix A). Our results show that PVJ1 was present in the original sediment sample (Appendix A). Members of the family Bacillaceae are known to be infected by bacteriophages. However, there were no reports of bacteriophages infecting the genus *Psychrobacillus* of this family. Therefore, PVJ1 is the first bacteriophage isolated from *Psychrobacillus*.

As revealed quantitatively by qPCR and observed by TEM, PVJ1 virions were released at a low frequency without induction under our experimental conditions (Figure 1D). A brief heat treatment at 60 °C or growth at pH4 resulted in a drop in the noninduced release of the phage (Appendix A). The PVJ1 DNA increased drastically following treatment with mitomycin C, which inhibited the growth of the host cells (Figure 1C,D). The phage production, which showed a latent period of ~4 h, peaked at 1.93 × 10^11^ copies/mL at 17 h after the addition of mitomycin C (Figure 1D).

The host range of PVJ1 was tested using plaque assays as well as qPCR on six different strains including *Psychrobacillus* sp. GC2J1 (CGMCC 1.19103), *Psychrobacillus insolitus* DSM5 (CGMCC 1.3683), *Psychrobacillus lasiicapitis* NEAU-3TGS17^T^ (CGMCC 1.15308), *Lysinibacillus antri* SYSU K30002^T^ (CGMCC 1.13504), *Paenisporosarcina antarctica* N-05 (CGMCC 1.6503), and *Planococcus rifietoensis* N14 (CGMCC 1.8017). VLPs were readily observed only in *Psychrobacillus* sp. GC2J1 after MMC induction and none of the other tested strains were infected by PVJ1.

### 3.2. General Features of the PVJ1 Genome

PVJ1 harbors a 53,187 bp linear dsDNA genome (see below) with a GC content of 37.43%, which is similar to that of the genome of *Psychrobacillus* sp. GC2J1 (37.1%). PVJ1 is predicted to encode 84 proteins, of which 77 have homologous sequences (E-value < 1 × 10^−3^) in the NCBI database and 39 have an annotating function (Appendix A). Genes encoding DNA replication, transcription, repair, and integration proteins (blue), proteins related to the host cell lysis (red), structural proteins (purple), and DNA packaging (brown) exist in cassettes in the genome (Figure 2). Most of the genes are arranged in the same orientation.

In the gene cassette responsible primarily for DNA replication, transcription, repair, and integration, ORF12 is a homolog of Holliday junction resolvase, an enzyme that both resolves the Holliday structures resulting from the recombination process and serves roles in debranching phage DNA structures prior to phage genome packaging into head particles and in degrading bacterial host DNA [31]. ORF29 encodes an AAA^+^ family protein [32]. Both ORF33 and ORF34 code for HTH domain-containing transcriptional regulatory factors of the xenobiotic response element (XRE) family. These transcriptional factors are often encoded by phages of spore-forming bacteria and may be related to the regulation of integrase [33]. ORF35 encodes a site-specific integrase. A 42-bp integration core sequence exists downstream of ORF35, allowing the phage DNA to be integrated at a specific site in the host genome. ORF45, ORF46, ORF47, and ORF49 encode a YolD-like family protein, DNA polymerase V, regulatory protein LexA, and DNA ligase, respectively, and may play key roles in phage DNA replication and repair.

The second cassette of genes appears to be involved in PVJ1 lysis. ORF50 encodes an N-acetylmuramoyl-L-alanine amidase, which acts on the cell wall of the host when the virions are released [34]. Holin encoded by ORF51 presumably disrupts the host cell wall once the phage entered the lytic cycle, releasing the progeny phage particles [35]. ORF56 encodes a peptidoglycan/LPS O-acetylase, which comprises acyltransferase and SGNH hydrolase domains [36]. ORF57 encodes a polygalacturonase, which is also involved in the lysis of the cell wall.

In the gene cassette for structure proteins, ORF65 encodes a tape measure protein (TMP). It has been suggested that TMP is involved in the thermal stability of some phages [37]. The tail core protein, encoded by ORF67, and the tail sheath protein, encoded by ORF68, are the major structural proteins of the phage tail. The head-to-tail linking protein, encoded by ORF73, connects the head and tail of the virion and presumably forms a central channel for the injection of the phage DNA. ORF75 encodes MCP, from which the phage head is constructed. ORF76 encodes a head decoration protein. ORF80 encodes a head morphological protein of the SPP1 family, in agreement with the finding that PVJ1 is closely related to Bacillus phage SPP1 in a phylogenetic analysis of TerL (see below).

In the fourth gene cassette, ORF82 and ORF83 encode the large and small subunits of the terminal enzyme, respectively, serving important roles in phage DNA packaging.

BLASTN analysis revealed no significant similarity of the phage PVJ1 genome to other sequences in the viral database. However, comparative proteomic analysis revealed that the structural proteins of PVJ1 were similar to proteins encoded by various Firmicutes and phages. By sequence searching with a cutoff of 70% identity, 50% coverage, and E-value of 1 × 10^−3^, we retrieved from public databases a total of 77 phage sequences (16 from the IMG/VR database and 61 from the NCBI database) encoding homologs of PVJ1 MCP (Appendix A). Overall, 98.7% of these homologs are from *Caudovirales* with Firmicutes, predominantly Bacillaceae (40.25%) and Paenibacillaceae (37.66%), as putative hosts. In agreement with the widespread distribution of Firmicutes, phages encoding an MCP sharing significant similarity (70% identical amino acids) with that of PVJ1 are detected in samples collected from marine and freshwater environments, terrestrial soil, plants, and animals, as well as bioreactors (Appendix A).

We then analyzed synteny between the genome of PVJ1 and those of the other nine prophages. All of the nine phage sequences, identified as intact prophages by PHASTER from the NCBI reference sequence database, encode MCPs, sharing ≥80% identity to that of PVJ1 at the amino acid sequence level. As shown in Figure 3, all of these phage genomes including that of PVJ1 exhibit similar modular organization, and gene cassettes for structure proteins are highly conserved among these prophages. A modular organization would facilitate viral genome recombination following horizontal gene transfer. Phage head proteins including MCP, SPP1 family phage head morphogenesis protein, and portal protein are conserved in all 10 prophages, with the sole exception that the phage head morphogenesis protein is absent in *Lysinibacillus macroides* DSM 54 prophage 876,855–942,549, suggesting that these phages have a similar head structure. Similarities in additional genes, such as those encoding functions involved in the genetic processes, were detected between PVJ1 and *Bacillus* sp. S/N-304-OC-R1 Contig 9 164,151–206,780 (prophage BS) and *Psychrobacillus* sp. FJAT-21963 super1 735,434–790,565 (prophage PF). For example, the AAA family ATPase of PVJ1 (ORF29) shows 85.52% and 50.79% identities with prophage BS and prophage PF, respectively, and the putative metallo-hydrolase YycJ of PVJ1 (ORF25) is 84.98% and 65.24% identical to prophage BS and prophage PF, respectively. Despite these similarities, PVJ1 differs significantly from other known bacteriophages in the genome sequence.

The structural proteins of PVJ1 were also identified by SDS–PAGE and subsequent MALDI–TOF mass spectrometry. Five protein bands were obtained, suggesting that they are present in relative abundance in the virion. The five proteins correspond to those encoded by ORF81, ORF68, ORF75, ORF67, and ORF76 (Figure 4). As expected, MCP, encoded by ORF75, was the most abundant of the five proteins. A head decoration protein, identified as the product of ORF76, migrated faster than expected from its theoretical molecular mass (12,390 Da), presumably as a result of a posttranslational modification.

### 3.3. The PVJ1 Genome Is Linear and Packaged by a Headful Strategy

To determine whether the PVJ1 genome was linear or circular in the virion, the phage DNA was extracted from the purified phage particles. Five pairs of primers were designed to generate overlapping fragments, which together would cover the entire PVJ1 genome. We found that PCR with each primer pair was able to generate a product of the expected length based on the assumption that the phage genome was circular (data not shown). Surprisingly, however, molecules of the PVJ1 genome appeared linear with an average size of ~17 μm, as would be expected from the 53 kb genome length under an atomic force microscope (Figure 5A).

To understand the above discrepancy, the genome DNA from the purified PVJ1 virions was analyzed by restriction digestion with SacI and BlpI. Substoichiometric fragments were observed in the restriction digest (Figure 5B). These fragments appear to have resulted from packaging initiation cleavage by the terminase large subunits at one end and restriction cleavage at the other end, and, therefore, are the *pac* fragments [22]. The site of packaging cleavage was found in the middle of the TerS gene by sequencing the restriction fragments (Figure 5C). The restriction pattern suggests that the PVJ1 genome is linear and is probably packaged by a headful mechanism [22]. Since TerL, often used in phage classification, permits the classification of phages by DNA-packaging strategies and DNA ends [38], we compared the TerL sequences of 33 viruses, including PVJ1, thermophilic *Geobacillus* viruses D6E, and GVE2 [22]. As shown in Figure 6, the TerL protein of PVJ1 belongs to the group including TerL from *Samonella* virus P22, which has been shown to employ a headful DNA packaging strategy [39]. It is worth noting that TerL from *Geobacillus* virus D6E, which was also isolated from a hydrothermal vent, shows a high similarity to that from PVJ1 at the amino acid sequence level (73%), suggesting that the two TerL proteins share a recent common ancestor. Our phylogenetic analysis supports the notion that the PVJ1 genome is packaged by a headful mechanism.

### 3.4. PVJ1 Integrates into the Host Genome at the Purine Nucleoside Phosphorylase Gene

To identify the site of integration of PVJ1, we sequenced the genome of *Psychrobacillus* sp. GC2J1. As shown in Figure 6, the *attL* and *attR* sites of PVJ1 are present at the ends of the integrated prophage sequence. The *attB* site contains a 42 bp core sequence (CTGGATTGCTGCTTCTAATGCAACAACGATCATGTCGTTGAA), which was duplicated upon PVJ1 integration (Figure 7). The core sequence occurs once in the GC2J1 genome and is located near the 3’-end of the host *deoD* gene, which encodes purine nucleoside phosphorylase (PNP). Interruption of the *deoD* gene by phage insertion was reported in *Escherichia coli* [40]. The *attP* site is situated between ORF35, the integrase gene, and ORF36, which encodes a hypothetical protein. Integration of the PVJ1 genome may not affect the function of host-encoded PNP because a new stop codon is generated so that the altered PNP polypeptide differs only slightly from the original one at the C-terminus (Figure 7). By comparison, the integration site of the thermophilic virus GVE3 is in the gene encoding a pyrimidine nucleoside phosphorylase [41]. The tRNA or tmRNA genes are known to be preferred sites of phage integration in prokaryotes [42]. Our observations suggest that host genes for nucleotide metabolism may also serve as a hot spot for phage integration. However, the addition of purine or purine nucleoside did not affect the growth of *Psychrobacillus* sp. GC2J1 or the replication of PVJ1 in the host cells (data not shown).

## 4. Conclusions

Hydrothermal vents, one of the most extreme environments on Earth, are abundantly inhabited by diverse microorganisms. However, our knowledge of the viral diversity in these unique ecosystems remains limited. In this article, we reported the isolation and characterization of PVJ1, the first-tailed temperate phage-infecting *Psychrobacillus*. PVJ1 possesses a 53,187 bp linear dsDNA genome. The phage packages DNA by a headful mechanism and integrates its genome into the host chromosome at the 3′-end of the *deoD* gene. Phages with an MCP highly similar (≥70% identical amino acids) to that of PVJ1 are widespread in various environments and primarily infect Firmicutes. These findings will facilitate the study of the biology of bacteriophages and their ecological roles in marine hydrothermal vents.

## Figures and Tables

**Figure 1 viruses-14-00108-f001:**
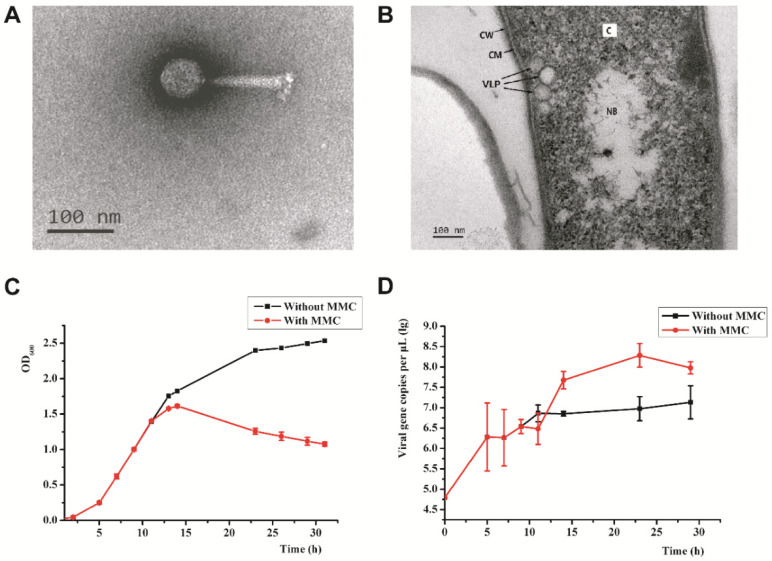
Morphology and induction of PVJ1: (**A**) transmission electron micrographs of PVJ1; (**B**) thin section of a *Psychrobacillus* sp. GC2J1 cell following treatment with mitomycin C. CM, cytoplasmic membrane; NB, nuclear body; C, cytoplasm; CW, cell wall; VLP, virus-like particle; (**C**) effect of mitomycin C treatment on the growth of *Psychrobacillus* sp. GC2J1; (**D**) titer of PVJ1 during mitomycin C induction experiment.

**Figure 2 viruses-14-00108-f002:**
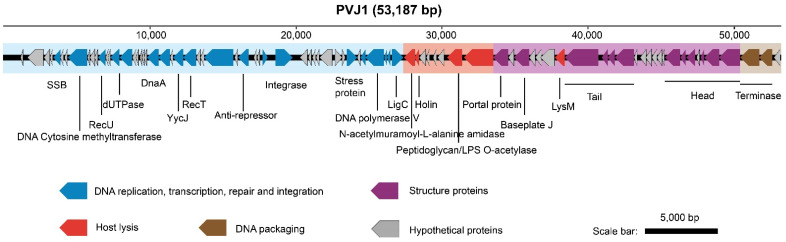
Genome organization of PVJ1. ORFs are shown with arrows oriented in the direction of transcription. Gene cassettes with putative functions are color coded as indicated.

**Figure 3 viruses-14-00108-f003:**
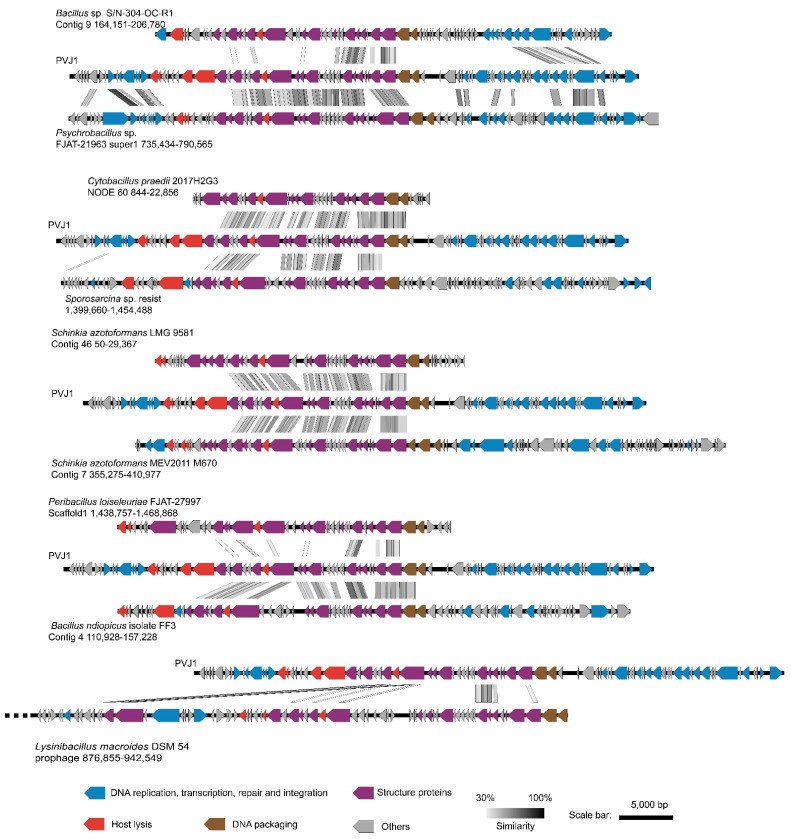
Genome comparison between PVJ1 and nine other prophages encoding an MCP highly similar to that of PVJ1. ORFs with putative functions are color coded as indicated. The nine prophage genomes used for synteny analysis include *Bacillus* sp. S/N-304-OC-R1 Contig 9 164,151–206,780, *Psychrobacillus* sp. FJAT-21963 super1 735,434–790,565, *Cytobacillus praedii* 2017H2G3 NODE 60,844–22,856, *Sporosarcina* sp. resist 1,399,660–1,454,488, *Schinkia azotoformans* LMG 9581 Contig 4650–29,367, *Schinkia azotoformans* MEV2011 M670 Contig 7 355,275–410,977, *Peribacillus loiseleuriae* FJAT–27997 Scaffold1 1,438,757–1,468,868, *Bacillus ndiopicus* isolate FF3 Contig 4 110,928–157,228, and *Lysinibacillus macroides* DSM 54 prophage 876,855–942,549.

**Figure 4 viruses-14-00108-f004:**
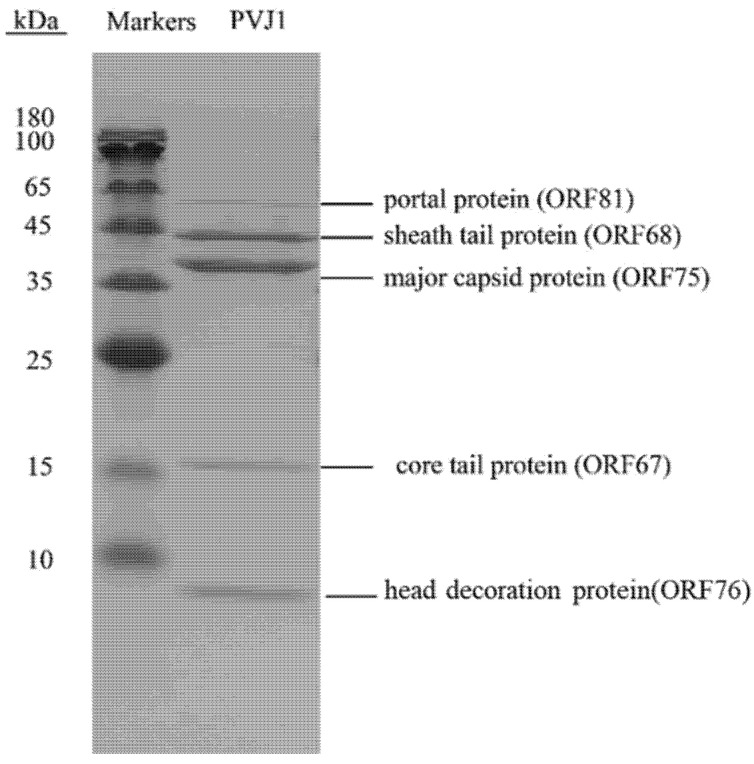
Structural proteins of PVJ1 as revealed by SDS–PAGE. A sample of purified PVJ1 virions was subjected to 12% SDS–PAGE. The gel was stained with Coomassie brilliant blue G250. Gel slices containing protein bands were excised. Proteins in the gel slices were digested with trypsin, and the resulting peptides were analyzed by MALDI–TOF mass spectrometry. The identity of the proteins is shown.

**Figure 5 viruses-14-00108-f005:**
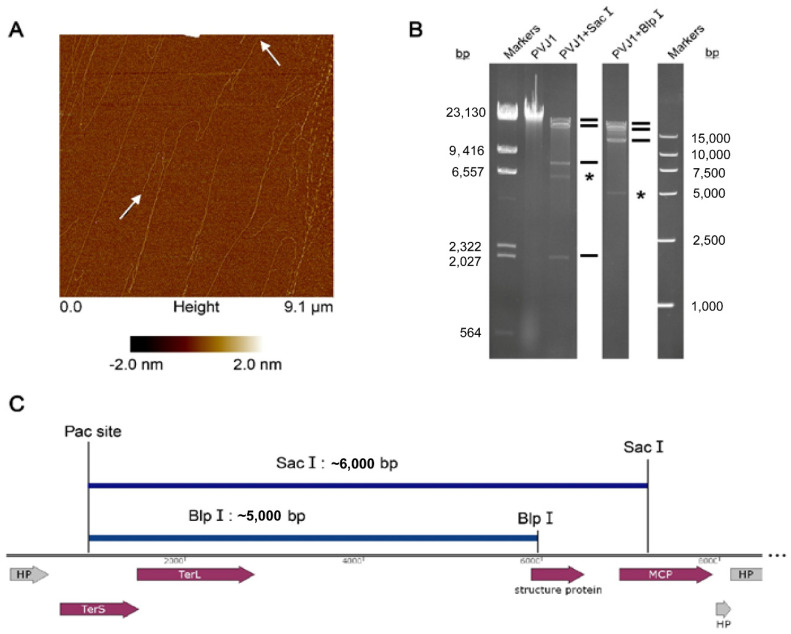
Analysis of the structure of the PVJ1 genome: (**A**) AFM images of the purified genomic DNA of PVJ1. White arrows point to the two ends of the linear genome; (**B**) restriction digestion of the PVJ1 DNA. The purified PVJ1 DNA was digested with either BlpI or SacI. The restriction fragments were subjected to electrophoresis in 0.8% agarose gel. The gel was stained with ethidium bromide and photographed under UV light. Expected restriction fragments are shown by thick horizontal lines. The *pac* fragments generated by SacI and BlpI cleavage are indicated with a star (*); (**C**) a diagram showing the sizes of the pac fragments generated through packaging initiation cleavage and restriction cleavage with SacI and BlpI.

**Figure 6 viruses-14-00108-f006:**
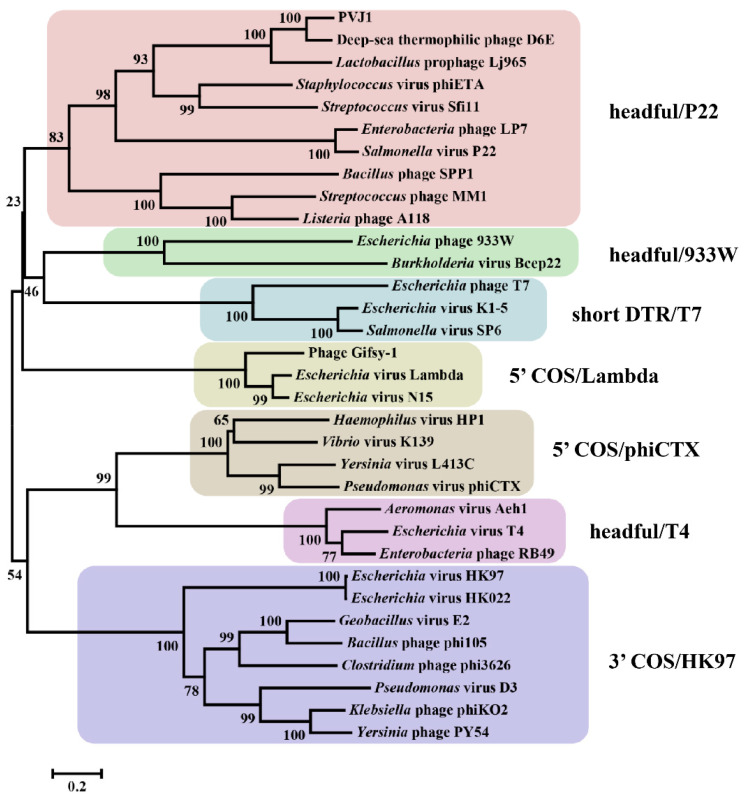
Phylogenetic analysis of the large terminase subunits of PVJ1 and selected phages. The phylogenetic tree was constructed by using the neighbor-joining method. Bootstrap analysis was performed with 1000 repetitions.

**Figure 7 viruses-14-00108-f007:**
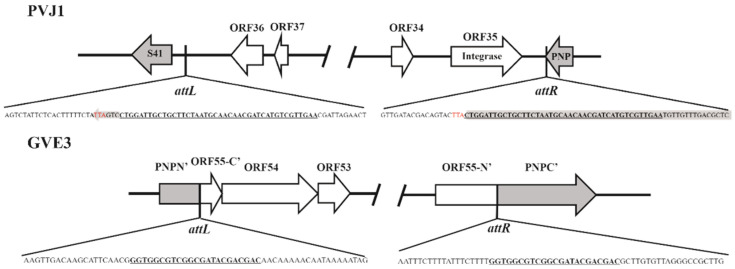
Sites of integration by PVJ1 and GVE3. Host genes at the site of phage integration in the host genome are shown with grey arrows. The *attL* and *attR* sequences are underlined. The stop codons are in red. The gene sequences encoding the purine nucleoside phosphorylase of *Psychrobacillus* sp. GC2J1 and pyrimidine nucleoside phosphorylase of *G. thermoglucosidasius* are in grey. PNP, purine nucleoside phosphorylase or pyrimidine nucleoside phosphorylase; PNPN’ and PNPC’ are the N- and C-terminal portions of the *G. thermoglucosidasius* pyrimidine nucleoside phosphorylase.

## Data Availability

The genome sequence of PVJ1 is available in the GenBank database under accession numbers MZ983385. The accession number of the host strain’s 16S rRNA gene sequence available in the GenBank database is MZ959810 (*Psychrobacillus* sp. GC2J1).

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
