# Peer review of "Isolation and Characterization of the First Temperate Virus Infecting Psychrobacillus from Marine Sediments"

_viruses, 2022, doi:10.3390/v14010108_

Round 1

Reviewer 1 Report

In this manuscript, Liu et al, isolated Psychrobacillus strain GC2J1 from marine sediment, and then they characterized a temperate phage PVJ1 which was integrated in the GC2J1 genome. Generally, the manuscript is well written and the analysis was well conducted and detailed. Currently, the isolation of bacteriophages from marine sediment, especially at a depth of deep ocean, is rare. This study provides important data and new insights for understanding the temperate phages inhabiting marine sediment. Despite these progresses, there are still some concerns, in my opinion, that need to be addressed. Specific comments are listed below.

Line 216-217. The author stated that ”PVJ1 was only capable of visibly infecting the host strain Psychrobacillus sp. GC2J1”. This result is abnormal, based on what I know, Shouldn't the host of a temperate phage be immune to the infection of the same phage?

Line 294-296 and Figure 4A, from my perspective, it is difficult to distinguish individual DNA molecules, and their respective ends, from the AFM image. Moreover, is it possible that the circular DNA molecule breaks into a linear type during the DNA extraction and AFM observation? How many DNA molecules were examined? Are they all linear? So overall, I think the evidence provided by AFM that the PVJ1 genome is linear is not convincing.

Page 10, Figure 7, I’m confusing about the phylogenetic tree. Was the tree constructed based on the amino acid sequence of MCPs? Or based on the concatenated sequence of MCP and terminase large subunit?

Page 11, Line 355, The author found “157 from the NCBI database”, I guess the author only searched NCBI RefSeq virus database, if so, I would suggest to include prokaryotic genomes in the sequence searching, as the PVJ1-like bacteriophages probably integrated in some genomes.

Page 6, Because the genomic content of PVJ1 was introduced by different gene cassettes, I would suggest that Figure 2A could be modified to indicate these different cassettes, thereby facilitating a better interpretation of the PVJ1 genome.

Line 162-168, “capsid proteins retrieved…”, It is more appropriated to move this part to “2.9 Phylogenetic analysis”.

Line 187-189, How many VLPs were analysed to get the data of “diameter, length, width”? This information is needed to be indicated in the manuscript.

Line 286, what do you mean “genetic mechanism”?

Line 70, “transportation.5g”, Missing spaces.

Reviewer 2 Report

The manuscript reports on the first bacteriophage isolated from Psychrobacillus. However, the use of ‘novel’ in the title is not justified, since the authors do not show where this phage stands among other phages/prophages, what unique features the new phage has, etc.  Blastp searches against GenBank nr database initiated with PVJ1 proteins bring multiple Bacillales hits, which indicates that phages highly similar to PVJ1 are common among Bacillales. It is advisable to include proteins closest to those of PVJ1 into the phylogenetic analysis presented in the paper (i.e., Figure 5, TerL). 

‘Biogeographic distribution and host range’ part merely reflects the distribution of Bacillales. The fault of this part is that it reports on the distribution of ‘PVJ1-like phages’ whereas the definition of ‘PVJ1-like’ clade is lacking.

Bioinformatic part of the study needs to be improved/redone.

Minor comments:

Line 24 ‘. Phages sharing structural proteins with PVJ1 are widespread in diverse environments <…> ‘ – this sentence might be omitted, since all tailed phages share structural proteins with PVJ1 (capsid and TerL), and tailed phages, indeed, are widespread in diverse environments.

Line 60 ‘PVJ1-like viruses’ – the definition of ‘PVJ1-like’ clade is lacking. This sentence should be removed.

Line 101 ‘Translated ORFs were annotated using BLASTp at NCBI with E-value ≤ e-3’  - while blastp detects proteins with some sequence similarity, it should not be used for protein functional annotation. Searches against conserved domain databases or solved structure databases should be used instead.

Line 148 ‘terminase large subunits of PVJ1 and 32 other viruses’ – it’s unclear how these 32 reference sequences were chosen.

Line 153 ‘Global distribution’ – both TerL and capsid proteins were collected; only capsid proteins were used for a tree; it’s unclear how the collected TerL sequences have been utilized.

Line 183 – ‘sp. GC1’ – perhaps the authors meant sp. GC2J1

Line 229 ‘Comparison of genomes of PVJ1 and two related phages’ – the choice of the two related phages is poorly justified (see Line 276-287 comment).

Lines 232-262 list functional annotations of some of the predicted ORFs; it would be more readable as a table than a text.

Line 268 ‘hypothetical protein, 267 identified as the product of ORF76’ is a head decoration protein (that can be detected with HHpred online tool, https://toolkit.tuebingen.mpg.de/tools/hhpred). This instance emphasizes the necessity of such tools for protein functional annotation.

Line 276-287 ‘Genome annotation by PHASTER shows that the genome of PVJ1 most closely resembles those of phage phiOH2 (NC_021784), which infects the thermophile Thermus, and phage Pony (NC_022770), which infects Bacillus.’ 

The use of PHASTER is not mentioned in Methods; PHASTER is not a genome annotation tool; ‘closely resembles’ lacks definition. This paragraph and the corresponding part of Figure 2 need to be removed.

Line 311 ‘virus D6E <…> shows the highest similarity to that from PVJ1 at the amino acid sequence level’ – this is true if the reference sequences are restricted by viruses. Numerous Bacillales genomes contain prophages highly similar to PVJ1 phage.

Lines 351-361, Biogeographic distribution – the whole part might be omitted.

Line 363, Figure 7 legend – two sequence collections are mentioned, TerL and MCP, and it’s unclear how the two were utilized to construct the tree.

How the tree is rooted?

Lines 369-374 belong to Introduction rather than to Conclusions

Line 377 again, claims regarding PVJ1-like phages should be removed.

Round 2

Reviewer 1 Report

The authors have addressed all my concerns, and I have no further comments.

Author Response

Dear Reviewer,

We gratefully thank you for your positive comments and the precious time you spent making our manuscript better!

Reviewer 2 Report

The manuscript has been substantially revised to comply with Reviewer recommendations. There is one more suggestion: throughout the manuscript, the questionable ‘PVJ1-like’ term has been replaced with ‘Phages with a major capsid protein highly similar to that of PVJ1’. For better clarity, it’s advisable to specify the percent of identity in parentheses, e.g., ‘major capsid protein highly similar (XX% identical amino acids) to that of PVJ1’.

Author Response

Dear Reviewer,

Thank you very much for your kindly comments on our manuscript. There is no doubt that these comments are valuable and very helpful for improving our manuscript.

As suggested, we have changed the words “Phages with a major capsid protein highly similar to that of PVJ1” to “major capsid protein highly similar (70% identical amino acids) to that of PVJ1” in the revised manuscript (p. 1, lines 25, p. 2, lines 60, p. 8, lines 279-280, and p. 11, lines 385).

Thank you again for your positive and constructive comments and suggestions on our manuscript. We hope you will find our revised manuscript acceptable for publication.